# THE EFFECTIVENESS OF RANDOM FORGETTING FOR ROBUST GENERALIZATION

**Vijaya Raghavan T Ramkumar**[1], **Bahram Zonooz**[*1,2] **& Elahe Arani**[* 1,3]
[1]Eindhoven University of Technology   [2]TomTom   [3]Wayve
raghavijay95@gmail.com, b.zonooz@tue.nl, e.arani@gmail.com

## ABSTRACT

Deep neural networks are susceptible to adversarial attacks, which can compromise their performance and accuracy. Adversarial Training (AT) has emerged as a popular approach for protecting neural networks against such attacks. However, a key challenge of AT is robust overfitting, where the network's robust performance on test data deteriorates with further training, thus hindering generalization. Motivated by the concept of active forgetting in the brain, we introduce a novel learning paradigm called "Forget to Mitigate Overfitting (FOMO)". FOMO alternates between the forgetting phase, which randomly forgets a subset of weights and regulates the model's information through weight reinitialization, and the relearning phase, which emphasizes learning generalizable features. Our experiments on benchmark datasets and adversarial attacks show that FOMO alleviates robust overfitting by significantly reducing the gap between the best and last robust test accuracy while improving the state-of-the-art robustness. Furthermore, FOMO provides a better trade-off between standard and robust accuracy, outperforming baseline adversarial methods. Finally, our framework is robust to AutoAttacks and increases generalization in many real-world scenarios.[1]

## 1 INTRODUCTION

Deep neural networks (DNNs) have demonstrated outstanding performance in various domains, ranging from computer vision to natural language processing and speech recognition. However, recent studies (Szegedy et al., 2013; Goodfellow et al., 2014) have revealed the susceptibility of DNNs to adversarial attacks. These attacks are initiated by adding small, yet intentionally crafted, imperceptible perturbations to input data, resulting in erroneous predictions by DNNs. The adversarial attacks pose a serious security threat in applications such as autonomous vehicles, medical diagnosis, and other areas where DNNs are used to make critical decisions. Adversarial Training (AT) (Madry et al., 2017; Zhang et al., 2019a) has emerged as a promising solution to address the issue of robustness and security of DNNs against adversarial attacks. It involves training DNNs using adversarial examples to improve their resilience against such attacks.

Recently, Rice et al. (2020) reported "robust overfitting" in AT, where the robust performance on test data degrades with further training. Figure 1 (left) illustrates this phenomenon, where the adversarial test accuracy significantly lags behind the adversarial train accuracy, leading to robust overfitting. Although this phenomenon is present in AT, conventional methods to prevent benign overfitting in standard training, such as explicit regularization and data augmentation, do not improve performance compared to the best accuracy achieved with early stopping in AT. While early stopping is a useful technique to prevent robust overfitting, it may not be desirable due to the occurrence of the "double descent"[2] phenomenon in AT (Rice et al., 2020; Nakkiran et al., 2021). Thus, the potential existence of robust overfitting in AT, and the failure of conventional methods to mitigate it, present a striking lacuna in building robust machine learning systems.

---

[*]Contributed equally.

[1]Code is available at https://github.com/NeurAI-Lab/FOMO.

[2]Double descent is a phenomenon in deep learning where a model's test error initially increases, decreases, and then increases again as model complexity or dataset size increases (Nakkiran et al., 2021).

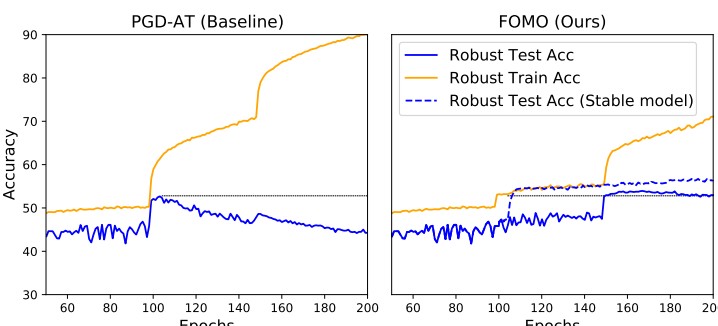

Figure 1: Comparison of robust overfitting in baseline PGD-AT (left) and our method FOMO (right), highlighting the gap between robust training and test accuracy on CIFAR-10 with PreAct-ResNet18. FOMO significantly reduces robust overfitting compared to the baseline's best early-stop checkpoint.

Unlike DNNs, humans excel in generalization in dynamic environments, facilitated by the interplay of remembering, forgetting, and relearning processes in the brain (Richards & Frankland, 2017). As the eminent psychologist William James noted, *"If we remembered everything, we should on most occasions be as ill off as if we remembered nothing."* Paradoxically, one condition for remembering is that we should forget. Similarly, our brain has the remarkable ability to remember and actively forget information as needed, which is necessary for learning and achieving generalization. Although the underlying mechanism of active forgetting remains elusive, neuroscience and cognitive psychology research (Gravitz, 2019; Izawa et al., 2019) provides growing evidence that the brain actively forgets by pruning neurons, shaping the learning-memory process (Shuai et al., 2010; Hardt et al., 2013; Davis & Zhong, 2017; Richards & Frankland, 2017). Furthermore, the interaction between forgetting, remembering, and relearning is reinforced by the presence of multiple memory systems. Consolidation through long-term memory storage enables us to preserve crucial knowledge for future use and retrieval, while relearning reinforces previously learned information (Bjork & Allen, 1970). Together, these three components work in harmony in the regulation of the learning process to achieve better generalizability in the real world. Therefore, emulating these aspects in DNNs might hold the key to achieving robust generalization in AT.

Therefore, we propose a general learning paradigm, which we refer to as *FOrgetting for Mitigating Overfitting (FOMO)*, to address the problem of robust overfitting of parameterized networks during AT. We consciously simulate the process of active forgetting in the DNNs by re-randomizing a random subset of weights periodically during AT. Each forgetting phase is followed by a relearning phase, which we call 'interleaved training'. Our method alternates between the forgetting and relearning phases while consolidating generalized features. With extensive experiments on multiple datasets, we show that our proposed training paradigm boosts the robust performance and generalization of AT models to a greater extent by alleviating robust overfitting. Our main contributions are as follows;

- **FOMO**, an adversarial training paradigm to improve the performance and generalization of DNNs through the lens of active forgetting and relearning.
- We demonstrate the efficacy of FOMO against the AutoAttacks.
- Our method alleviates robust overfitting and achieves significant results across multiple architectures and datasets.
- Our proposed training paradigm is robust to natural corruptions and leads to flatter minima.

## 2 RELATED WORK

### 2.1 ADVERSARIAL LEARNING

Adversarial training (AT) has been shown to be an effective method of countering adversarial attacks; thus we define adversarial learning settings as envisioned in Madry et al. (2017). We consider a classification task with a given K-class dataset $D = \{(x_i, y_i)\}_{i=1}^n \subseteq X \times Y$, where $x \in \mathbb{R}^d$ represents an input sampled from a certain data-generating distribution $P$ in an i.i.d. manner, and $Y := 1, \ldots, K$ represents a set of possible class labels. Let $f_\theta \in \mathbb{R}^d$ be a neural network modeled to predict classes. The notion of adversarial robustness requires $f_\theta$ to perform well not only on $P$ but also on the worst-case distribution near $P$ under a certain distance metric. More concretely, the

adversarial robustness we primarily focus on in this paper is the $\ell_p$-robustness: that is, for a given $p \geq 1$ and a small $\epsilon > 0$, we aim to train a classifier $f_\theta$ that correctly classifies $x + \delta$ for any $||\delta||_p \leq \epsilon$ as $y$, where $(x, y) \sim P$.

In AT, the training data is sampled from adversarial regions incorporated to train the classifier. Madry et al. (2017) formulated AT as a min-max optimization problem:

$$\min_\theta \sum_{i=1}^n \max_{\|\delta\|_p \leq \epsilon} \mathcal{L}_{adv}(f_\theta(x_i + \delta), y_i), \tag{1}$$

where $f_\theta$ is the DNN with parameters $\theta$, and $\mathcal{L}_{adv}(.)$ is the typical classification loss function (e.g., the cross-entropy (CE) loss). The inner maximization is to find an adversarial example $x_0^i$ that maximizes the loss. The outer minimization is to optimize network parameters $\theta$ that minimize the loss on adversarial examples.

## 2.2 ROBUST GENERALIZATION.

Unlike in standard training, where longer training results in near-zero training and test error, Rice et al. (2020) observed that the test error increases in adversarial training. This phenomenon is called robust overfitting. Schmidt et al. (2018) established that achieving an adversarially robust generalization is challenging and requires more training data. Zhang et al. (2019b) pointed out the limitations of adversarial training to blind spot attacks. Subsequently, several empirical approaches have been proposed to improve generalization, such as adversarial training with semi/unsupervised learning (Carmon et al., 2019; Zhai et al., 2019; Zhang & Xu, 2019), AVmixup (Lee et al., 2020), and robust local feature (Chen et al., 2021). However, we distance ourselves from these works, as these data interpolation methods rely heavily on the requirement of large datasets to mitigate overfitting.

In contrast, Rice et al. (2020) systematically investigated various techniques used in deep learning, including $\ell_1$ and $\ell_2$ regularization, cutout, mixup, and early stopping, and found that early stopping was the most effective approach to remedy robust overfitting. While early stopping may not be the optimal solution for robust overfitting, other approaches mitigate this by promoting model flatness (Wu et al., 2020; Chen et al., 2020). Chen et al. (2020) achieves model flatness by leveraging knowledge distillation to smooth the logits space while applying stochastic weight averaging (Izmailov et al., 2018) to smoothen the weights space. However, their method is computationally expensive as it necessitates pre-training additional models to mitigate overfitting. Dong et al. (2021) incorporates the temporal ensembling (TE) technique (Laine & Aila, 2016) into the AT frameworks to regularize the predictions of adversarial examples from becoming overconfident. On the other hand, Adversarial Weight Perturbation (AWP) (Wu et al., 2020) explicitly regularizes the flatness of the weight loss landscape by proposing a double-perturbation mechanism that adversarially perturbs both inputs and weights. However, AWP requires an additional maximization step to compute the adversarial noise to perturb the network weights during AT. IDBH (Li & Spratling, 2023) underscores the importance of diversity and hardness in data augmentation, showing that diversity improves accuracy and robustness, while hardness enhances robustness in adversarial training. Unlike the other approaches, we tackle the problem of robust overfitting in AT through the lens of active forgetting. Given the ability of DNNs to memorize noise in the training data (Dong et al., 2021), we hypothesize that actively forgetting and relearning during AT may help consolidate generalizable features and achieve robust generalization.

## 2.3 BIOLOGICAL UNDERPINNINGS FOR ROBUST GENERALIZATION

We begin by motivating our approach through an examination of learning dynamics within the human brain. Intelligent decision-making in noisy, dynamic environments emerges from the interplay between memory retention and forgetting mechanisms. As highlighted by Davis & Zhong (2017), the ability of humans to generalize from new experiences is, in part, attributable to the phenomenon of active forgetting. Active forgetting plays a pivotal role in the selective regulation and rebalancing of the learning-memory process, thereby guarding against overfitting to individual experiences (Gravitz, 2019). These insights from neuroscience provide substantial evidence for the existence of a symbiotic relationship between generalization and active forgetting in biological neural networks, a relationship that remains notably absent in DNNs.

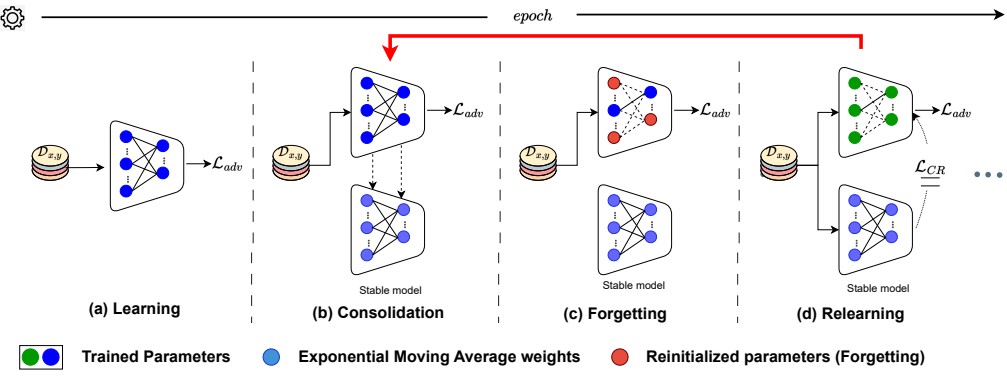

Figure 2: Schematics of the proposed *FOMO* framework illustrating its three pivotal phases during the AT. Beginning with standard learning, FOMO sequentially incorporates consolidation, a unique forgetting phase, and a relearning stage. This cyclic process enhances the robustness of the model by addressing adversarial overfitting through active forgetting and relearning.

In line with this, Richards & Frankland (2017) present compelling evidence supporting the essential role of forgetting in facilitating adaptive behavior within dynamic settings. Active forgetting within memory models offers functional advantages for robust generalization in such contexts, including (1) bolstering behavioral flexibility by diminishing the impact of obsolete information on memory-driven decision-making, and (2) averting overfitting to past events, thereby fostering generalization. Consequently, we underscore the importance of incorporating active forgetting mechanisms into computational models to effectively mitigate robust overfitting and promote robust generalization.

## 3 METHODOLOGY

We propose *FOrget to Mitigate Overfitting (FOMO)*, a training paradigm to improve generalization in adversarial learning. FOMO interchanges between the consolidation, forgetting, and relearning phases. More details can be found in Figure 2 and in Algorithm 1 in Appendix

### 3.1 FORGETTING

**What is forgetting?** We define the "forgetting step" as any process that results in a reduction in robust training accuracy. Specifically, let us denote the training dataset as $D$ and a neural network parameterized by $\theta$ as $f_\theta$. The robust training accuracy of the network $f_\theta$ is expressed as $Acc_R(f_\theta) = \frac{1}{n} \sum_{i=1}^{n} \mathbb{I} f_\theta(x_i + \delta) = y_i$. Now, consider the robust chance accuracy denoted as $C$ for a randomly initialized neural network ($f'_\theta$) on the dataset $D$. We define a forgetting step as any function satisfying two conditions: (i) $P(C < Acc_R(f'\theta) < Acc_R(f\theta)) = 1$, ensuring that the forgetting stage allows relearning due to a decrease in accuracy, and (ii) mutual information $I(f_\theta; D) > 0$ for the network concerning the dataset remains positive after the forgetting step. This concept captures the notion that forgetting entails the partial removal of information from the network, not complete erasure. Put simply, after any forgetting operation, the training accuracy of the adversarially trained network should fall between the chance accuracy and the robust accuracy achieved before forgetting. Thus, the forgetting step aims to strike a balance between preserving essential information from the previous step and facilitating relearning.

**Where to forget?** Now that we have established the process of forgetting, the subsequent question arises: Where in the neural architecture should forgetting take place to achieve generalization? Although our goal is to mitigate overfitting in AT, our aim is to prioritize characteristics that are simple, general, and beneficial for generalization (Valle-Perez et al., 2018). Therefore, we limit the forgetting process to the later layers of the network. For example, in the case of PreAct-ResNet18 (He et al., 2016), the last two layers are considered as later layers. This is because these layers have more capacity to memorize and tend to overfit the training data, while the earlier layers in the network tend to learn more generalized representations (Dong et al., 2021; Neyshabur et al., 2018;

Arpit et al., 2017; Yosinski et al., 2014; Zhang et al., 2021; Geirhos et al., 2018). Furthermore, studies in AT such as (Bakiskan et al., 2022; Siddiqui & Breuel, 2021) have shown that the network depth has a significant impact on adversarial robustness, and early layers tend to contribute more to robustness than later layers. Therefore, by limiting the forgetting process to the last two layers, we aim to regularize the weights in the later layers while retaining more generalized features learned in the earlier layers to facilitate robust generalization.

**How to forget?** Having established the concept of forgetting and identified where in a neural network to apply forgetting, the next critical step is to determine how to effectively emulate the forgetting process in DNNs. To this end, we consider a deep network ($f_\theta$) with $l$ layers, each with parameters $\theta_l$. Initially, the network undergoes adversarial training during a warm-up period, employing the adversarial example generation technique outlined in Section 2.1.

Our forgetting approach starts by splitting the network $f_\theta$ into two hypotheses during the AT as outlined by a binary mask $M$: (a) Retain hypothesis $F_\triangle$, and (b) Reset hypothesis $F_\triangledown$. The network parameters $\theta_l$ belonging to the retain and reset hypotheses are selected randomly using a binary mask $M_l$ such that $sum(M_l) = s * |\theta_l|$, where $s$ is the sparsity rate that determines the percentage of parameters belonging to the reset hypothesis $F_\triangledown$. We prefer random selection as it is simple and efficient. Note that the forgetting step applies only to the parameters of the later layers defined by the layer threshold $L$. Therefore, by default, the parameters $\theta_l$ corresponding to the early layers of the network belong to the retain hypothesis $F_\triangle$ throughout the AT. Finally, these hypotheses are outlined by a binary mask; 1 for $F_\triangle$ and 0 for $F_\triangledown$, that is, $F_\triangle = M \odot f_\theta$ and $F_\triangledown = (1 - M) \odot f_\theta$.

After randomly selecting the subset of weights belonging to the retain and reset hypothesis, the parameters belonging to $F_\triangle$ from the previous learning step are retained, while $F_\triangledown$ is reinitialized or reset to a random value sampled from a uniform distribution. Formally, we reinitialize parameters in the layer, as follows $\theta_l = M_l \odot \theta_l + (1 - M_l) \odot \theta_{r_l}$, where $\theta_{r_l}$ is a randomly initialized tensor. Thus, we emulate the aspect of active forgetting through random reinitialization of the connections and regularize the parameters in the later layers throughout the AT to achieve robust generalization.

Finally, the new network, after forgetting, contains an amalgamation of retained and reinitialized parameters that undergo relearning until the onset of the next forgetting step.

### 3.2 CONSOLIDATION

Since our FOMO method alternates between the forgetting and relearning phases during the AT, it is important to assimilate the generalized information that is relearned after each forgetting step. As the information is encoded in the parameters of the network (Krishnan et al., 2019), we intend to consolidate this information after each relearning step by employing another network $f_\phi$ called stable model similar to $f_\theta$. The knowledge learned by $f_\theta$ is consolidated in the stable model after each learning session (before each forgetting step), thereby serving as a long-term memory. This newly learned information is consolidated in the stable model by taking an exponential moving average of the $f_\theta$ weights with decay parameter $\alpha_c$: $\phi = \alpha_c \phi + (1 - \alpha_c)\theta$, where $\phi$ and $\theta$ correspond to the weights of the stable and the current network, respectively. It should be noted that the stable model is not subjected to training, while the forgetting operation is exclusively applied to $f_\theta$. Thus, this consolidation step can be considered as forming a self-ensemble of the intermediate model states obtained after multiple relearning steps that leads to generalized representations.

### 3.3 RELEARNING

Recent studies in cognitive neuroscience provide significant evidence of the existence of a symbiotic relationship between learning and forgetting (Gravitz, 2019; Richards & Frankland, 2017). The "spacing effect" is a well-known example of this relationship, which shows that long-term recall is improved when learning sessions are spaced out rather than massed together (Bjork & Bjork, 2019). Also, Bjork & Allen (1970) showed that the reduced accessibility of information between learning sessions is the key to regulating the important information in long-term memory. Therefore, we exploit this symbiotic relationship in AT by introducing an interleaved training session after each forgetting step. The network with new initialization undergoes a relearning phase wherein it is trained adversarially for $e_r$ epochs, where $e_r$ is less than the total number of AT epochs.

Table 1: Performance comparison on the CIFAR-10 using the PreActResNet-18 and WideResNet-34-10 architectures under a perturbation norm of $\epsilon_\infty = 8/255$.

| Method | PreActResNet-18 | | | | | | | WideResNet-34-10 | | | | | | |
| | Natural | | | PGD-20 | | | Trade-off | Natural | | | PGD-20 | | | Trade-off |
| | Best | Last | $\Delta$ | Best | Last | $\Delta$ | | Best | Last | $\Delta$ | Best | Last | $\Delta$ | |
|---|---|---|---|---|---|---|---|---|---|---|---|---|---|---|
| PGD-AT | 82.08 | 83.98 | 1.90 | 52.32 | 44.44 | -7.88 | 58.12 | 86.90 | 86.38 | -0.52 | 56.45 | 48.16 | -8.29 | 61.84 |
| TRADES | 80.72 | 82.61 | 1.89 | 52.66 | 49.75 | -2.91 | 62.10 | 84.73 | 84.62 | -0.11 | 56.50 | 47.28 | -9.22 | 60.66 |
| KD+SWA | **83.82** | **84.43** | 0.61 | 54.59 | 54.42 | **-0.17** | 66.18 | 86.85 | **88.03** | 1.18 | 56.92 | 55.74 | -1.18 | 68.25 |
| PGD-AT+TE | 82.15 | 82.59 | 0.44 | 55.03 | 53.79 | -1.24 | 65.14 | 86.20 | 85.63 | -0.57 | 56.89 | 53.49 | -3.4 | 65.84 |
| AWP | 81.25 | 81.56 | **0.21** | 55.39 | 54.73 | -0.66 | 65.50 | 86.28 | 86.27 | **-0.01** | 58.85 | 58.76 | **-0.09** | 69.90 |
| FOMO (Ours) | 81.84 | 82.51 | 0.67 | **56.68** | **56.46** | -0.22 | **67.04** | **87.31** | 87.08 | -0.23 | **59.69** | **59.23** | -0.46 | **70.50** |

Table 2: Performance comparison on CIFAR-100 and SVHN datasets, using the PreActResNet18 architecture and a perturbation norm of $\epsilon_\infty = 8/255$.

| Method | CIFAR-100 | | | | | | | SVHN | | | | | | |
| | Natural | | | PGD-20 | | | Trade-off | Natural | | | PGD-20 | | | Trade-off |
| | Best | Last | $\Delta$ | Best | Last | $\Delta$ | | Best | Last | $\Delta$ | Best | Last | $\Delta$ | |
|---|---|---|---|---|---|---|---|---|---|---|---|---|---|---|
| PGD-AT | 55.52 | 57.35 | 1.83 | 27.22 | 20.82 | -6.4 | 30.54 | 87.93 | 89.90 | -1.93 | 52.60 | 45.13 | -7.47 | 60.09 |
| TRADES | 55.53 | 57.09 | -1.56 | 29.56 | 26.08 | -3.48 | 35.80 | 90.88 | 91.30 | **0.42** | 52.50 | 47.50 | -5.00 | 62.48 |
| KD+SWA | 57.23 | **57.66** | 0.43 | 30.06 | 30.02 | -0.04 | 39.48 | 90.40 | 91.70 | 1.30 | 53.65 | 50.65 | -3.00 | 65.25 |
| PGD-AT+TE | 56.52 | 57.30 | 0.78 | 31.23 | 29.25 | -0.98 | 38.72 | 90.09 | 90.91 | -0.82 | 54.85 | 52.18 | -2.67 | 66.30 |
| AWP | 53.92 | 54.81 | -0.89 | 30.70 | 30.28 | -0.42 | 39.00 | 93.85 | 92.59 | -1.26 | 59.12 | 55.87 | -3.25 | 69.68 |
| FOMO | **57.45** | 57.07 | **-0.38** | **32.07** | **31.67** | -0.40 | **40.73** | **94.17** | **93.66** | -0.51 | **59.63** | **59.06** | **-0.57** | **72.44** |

**Regularization.** To expedite the relearning process, we propose a consistency loss that regularizes the output of the stable model. As shown in Equation 2, the function of this consistency regularization is to provide guidance to $f_\theta$ after each forgetting step. $\lambda_1$ and $\lambda_2$ in the Equation 2 denotes loss balancing parameters. The network ($f_\theta$) is updated so that it acquires new knowledge while aligning its decision boundary with the stable model that contains information consolidated across multiple relearning steps. This further prevents $f_\theta$ from robust overfitting during the relearning phase. The overall loss $\mathcal{L}_{FOMO}$ used to train the network during relearning phase is shown in Equation 3. We also detach the gradients from the stable model. Thus this regularization provides more detailed supervision from a stable model than $\mathcal{L}_{CE}$ loss, which helps to avoid overfitting. Once the network completes the relearning phase, the newly acquired knowledge is integrated into the stable model, which then proceeds to the forgetting step.

$$\mathcal{L}_{CR} = \lambda_1 \cdot D_{KL}(f_\theta(x)||f_\phi(x)) + \lambda_2 \cdot D_{KL}(f_\theta(x+\delta)||f_\phi(x+\delta)) \tag{2}$$

$$\mathcal{L}_{FOMO}(x, y; \theta) = \mathcal{L}_{adv}(f_\theta(x+\delta), y) + \mathcal{L}_{CR} \tag{3}$$

Thus, by cycling through forgetting, relearning, and consolidation, we introduce behavioral flexibility that helps in learning generalized information. Finally, for inference, we rely on the stable model, which serves as a long-term memory and holds the generalized knowledge that is consolidated after multiple relearning steps during the AT.

## 4 TACKLING ROBUST OVERFITTING

We compared our proposed method against previous AT methods on the CIFAR-10 dataset, utilizing two popular backbone networks, namely PreAct-ResNet18 (He et al., 2016) and WideResNet-34-10 (Zagoruyko & Komodakis, 2016). The performance of each approach was evaluated on the test set against the PGD-20 attack. As presented in Table 1, our experimental results demonstrate that FOMO outperforms all the baseline approaches regarding the best and last robust test accuracy for all architectures. Specifically, our method achieves a significant improvement of 12.02% and 11.07% in last-epoch robust test accuracy over PGD-AT on PreAct-ResNet18 and WideResNet-34-10, respectively. Notably, the last-epoch robustness of FOMO even consistently outperforms the best-epoch robustness of previous approaches that employ early stopping. Thus, early stopping may not be necessary, saving computation that goes into validating every epoch during AT. Moreover, FOMO achieves a better trade-off between standard and robust generalization than many AT methods. Lastly, FOMO mitigates robust overfitting by reducing the gap between the best and last

Table 3: White-box/Black-box (Auto-attack) performance comparison on CIFAR-10 and CIFAR-100, using the PreActResNet-18 architecture and a perturbation norm of $\epsilon_\infty = 8/255$.

| | Method | CIFAR-10 | | | CIFAR-100 | | |
|---|---|---|---|---|---|---|---|
| | | Best | Last | $\Delta$ | Best | Last | $\Delta$ |
| ICLR'18 | PGD-AT | 47.72 | 42.60 | -5.12 | 24.53 | 20.21 | -4.32 |
| ICML'19 | TRADES | 48.37 | 46.94 | -1.43 | 24.51 | 22.86 | -1.65 |
| NeurIPS'20 | AWP | 50.34 | 49.64 | -0.70 | 25.26 | 25.07 | -0.19 |
| ICLR'21 | KD+SWA | 49.87 | 49.74 | -0.13 | 26.04 | 25.99 | **-0.05** |
| ICLR'22 | PGD-AT+TE | 50.11 | 49.14 | -0.97 | 26.04 | 25.13 | -0.91 |
| ICML'22 | MLCAT$_{WP}$ | 50.70 | 50.32 | -0.38 | 25.86 | 25.18 | -0.68 |
| ICLR'23 | IDBH[Strong] | 50.74 | 49.99 | -0.75 | - | - | - |
| ICLR'24 | FOMO | **51.37** | **51.28** | **-0.09** | **27.57** | **27.49** | -0.08 |

robust test accuracy. Therefore, iterating between forgetting, relearning, and consolidation learns the generalized features that facilitate robust generalization in AT.

**Performance across different datasets.** To further evaluate the scalability of our proposed method, we conducted experiments on two additional benchmark datasets, CIFAR-100 and SVHN, which are more complex than CIFAR-10. The results, presented in Table 2, indicate that our method achieves the highest level of robustness in both the best and the final epoch, demonstrating its ability to effectively scale to larger datasets compared to other baselines. Thus, selectively forgetting information periodically through weight reinitialization regulates the weights and brings discernable benefits to the model regarding robust generalization.

## 5 TRAINING ROBUST ACCURACY

Here, we study the impact of FOMO on robust training accuracy. The variation in robust accuracy on the training data during AT is shown in Figure 1. Our proposed method (FOMO) effectively suppresses the robust training accuracy from the level attained by PGD-AT (from 92% to 69%). This illustrates that randomly forgetting a percentage of parameters in the later layers during AT inhibits the model from overfitting to the training data, leading to a significant reduction in the robust generalization gap (from 47.56% to approximately 12.54%) and, thus, mitigating robust overfitting.

## 6 EVALUATION WITH AUTOATTACK

DNNs are deployed in real-world settings where they face more realistic and challenging scenarios, including sophisticated attacks that can target the model. It is, therefore, essential to evaluate proposed approaches against strong attacks that can effectively compromise the model's robustness. By doing so, we can ensure that the adversarial methods can effectively enhance the model's security and resilience in real-world settings. Recently, AutoAttack (Croce & Hein, 2020) has been more effective in uncovering vulnerabilities in DNNs, making it a popular choice for evaluating the robustness of models. It uses an ensemble of several state-of-the-art white-box and black-box attacks to generate adversarial examples that are more transferable and harder to defend against. As shown in Table 3, we evaluate our proposed method against the AutoAttack on CIFAR-10 and CIFAR-100 datasets with the PreActResNet-18 architecture. Compared to the standard AT (PGD-AT) and other robust generalization methods, our approach largely alleviates robust overfitting under Auto-Attack on both datasets. These results indicate that training with FOMO leads to robust, consistently generalizable features across challenging adversarial attacks.

## 7 ROBUSTNESS AGAINST COMMON CORRUPTIONS

DNNs are commonly deployed in real-world settings, where they are exposed to dynamic environments influenced by factors such as lighting and weather. As a result, it is crucial to assess the robustness of DNNs to handle data distributions susceptible to natural corruption in real-world settings. Here, we evaluated the efficacy of our method on the corrupted CIFAR-10 dataset (Hendrycks

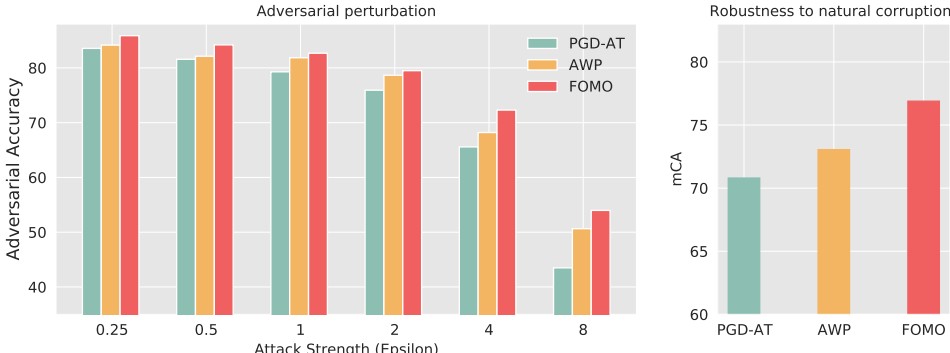

Figure 3: (Left) Robustness to adversarial attacks; (right) Robustness to Natural corruptions. In both robustness analyses, FOMO shows a significant performance improvement compared to the baselines considered.

& Dietterich, 2019), which includes 19 types of corruptions. Models are trained on clean images and tested on CIFAR-10-C. The mean corruption accuracy (mCA) of each method is presented in Figure 3 (Right). The results reveal that the mCA consistently improves with FOMO compared to PGD-AT across corruptions. Periodically iterating between forgetting, relearning, and consolidation during AT brings discernible benefits regarding robustness to natural corruptions.

## 8 ROBUSTNESS TO INCREASE IN ADVERSARIAL ATTACK STRENGTH

To further assess the efficacy, we conducted experiments employing a PGD-20 attack with perturbation strengths incrementally spanning from an $\epsilon$ value of 0.25/255 to 8/255. The outcomes, depicted in Figure 3 (Left), reveal a pronounced decline in PGD-AT's robustness as perturbation strength increases, whereas FOMO consistently surpasses baseline adversarial techniques across all strength levels, showcasing its stable performance. These results suggest that the process of forgetting and relearning within the FOMO framework consolidates high-level abstractions capable of withstanding minor data perturbations, in contrast to standard AT (PGD-AT) and AWP methodologies.

## 9 CONVERGENCE TO FLATTER MINIMA

DNNs that converge to flatter minima in the loss landscape demonstrate superior generalization, according to Neyshabur et al. (2018). Additionally, models that reside in flatter minima are more resilient since slight perturbations do not significantly affect their predictions. To evaluate the robustness of our method, we incorporate independent Gaussian noise into all parameters of the trained CIFAR-10 model, as outlined in Alabdulmohsin et al. (2021).

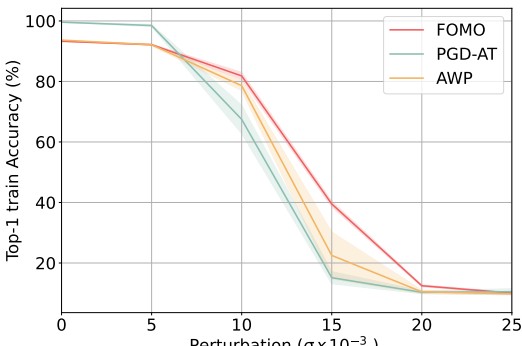

Figure 4 illustrates that the solution achieved by FOMO exhibits greater robustness to model perturbations compared to standard AT. Our method demonstrates significantly reduced sensitivity to perturbations. Specifically, for any level of noise introduced into the model parameters $\theta$, FOMO training accuracy exhibits smaller variations than standard training, suggesting that the FOMO solution may reside within flatter local minima. We posit that the training regimen involving alternating stages of for-

Figure 4: Robustness of the model perturbed by varying degrees of Gaussian noise. Our method is considerably robust to Gaussian perturbations, as the decline in performance is gradual, suggesting convergence to flatter minima.

Table 4: Ablation study of our proposed method on the CIFAR-10 dataset using PreActResNet-18 with a perturbation norm of $\epsilon_\infty = 8/255$. The numbers in green represent the relative gain of each step compared to the previous one.

| Method | PGD-20 | | | | AA | | | |
|---|---|---|---|---|---|---|---|---|
| | Best | Last | $\Delta$ | Rel. Gain | Best | Last | $\Delta$ | Rel. Gain |
| PGD-AT | 52.32 | 44.44 | -7.88 | - | 47.72 | 42.60 | -5.12 | - |
| + Forgetting | 52.32 | 48.86 | -3.46 | ↑56.12% | 47.72 | 44.63 | -3.09 | ↑39.65% |
| + Consolidation | 55.41 | 52.80 | -2.61 | ↑24.57% | 50.89 | 49.30 | -1.59 | ↑48.54% |
| + Regularization | **56.68** | **56.46** | **-0.22** | ↑91.57% | **51.37** | **51.28** | **-0.09** | ↑94.34% |

getting and relearning induces a broader valley, potentially elucidating our model's capacity to consolidate generalizable features.

## 10 ABLATION

We analyze the effect of forgetting, consolidation, and consistency regularization on the overall performance by incrementally adding each component to the baseline AT (PGD-AT). Table 4 reports the robust performance of the PreActResNet-18 network against PGD-20 and AA on the CIFAR-10 dataset.

Firstly, we evaluate the effect of forgetting by comparing the performance of our method with and without forgetting (PGD-AT). We periodically forget 3.5% parameters corresponding to the parameters in the later layers ( 3 and 4 in PreAct-ResNet18) every five epochs. Our observations indicate a substantial reduction in the gap between the best and last robust test accuracy when the forgetting component is incorporated, compared to PGD-AT. Introducing the forgetting component aids in the regularization of the weights and facilitates the release of the network's capacity to learn generalized features during relearning.

Furthermore, we extend our investigation by introducing a stable model for consolidating the features. Our findings indicate a notable improvement (approximately 3%) in the best test robust accuracy compared to PGD-AT. This improvement is attributed to the ability of the stable model to consolidate critical information learned during each relearning phase.

Lastly, we assess the role of the consistency regularization loss on the AT. Our observations reveal that the loss component is vital in mitigating robust overfitting by decreasing the gap between the best and last accuracy to -0.22. Thus, consistency regularization guides the network after each forgetting step by providing more precise supervision from a stable model, which helps to prevent overfitting to training labels. This further incentivizes the learning model to acquire generalized features consolidated in the stable model at the end of each relearning phase, resulting in an improved final performance. Our ablation study indicates that all individual components included in our proposed method are essential for robust generalization.

## 11 CONCLUSION

We introduce *Forget to Mitigate Overfitting (FOMO)*, an adversarial training paradigm, to improve DNN performance and generalization through the lens of active forgetting. FOMO alternates between the forgetting phase, which periodically forgets undesirable information in the model through the reinitialization of weights, and the relearning phase, which emphasizes learning generalizable features. These features are later consolidated into a stable model after each relearning phase. Empirical results show that the proposed framework improves both standard and robust performance and generalization across a wide range of architectures, datasets, and perturbation types. Our framework is robust to auto attacks and increases generalization in many real-world scenarios. Overall, FOMO presents a promising solution for achieving better robust generalization in adversarial training. In our future work, we will aim to further develop a theoretical understanding of this issue and how it relates to the effectiveness of AT, as the underlying cause of this robust overfitting phenomenon is not yet fully understood.

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

# A   APPENDIX

---

**Algorithm 1** Adversarial Training with FOMO

---

**Require:** Training data $D = (x_i, y_i)i = 1^n$, number of training epochs $N$, batch size $\mathcal{B}$, base adversarial training algorithm $\mathcal{A}$ with $\epsilon$, perturbation norm and steps, network $f_\theta$, stable model $f_\phi$, warm-up epochs $e_{warm-up}$, number of epochs the network undergoes relearning $e_r$, decay parameter $\alpha_c$, $s$ sparsity rate that determines the percentage of parameters to be forgotten in the later layers defined by layer threshold $L$.

1: Initialize model parameters $\theta$
2: **for** $epoch \leftarrow 1$ to $N$ **do**
3:     **if** $epoch > e_{warm-up}$ and $epoch \% e_r == 0$ **then**
4:         Consolidate($f_\theta$, $f_\phi$, $\alpha_c$)
5:         Random_forgetting($f_\theta$, $s$)       ▷ Randomly reinitialize the parameters in the later layers
6:     Sample a mini-batch $\mathcal{B} = (x_i, y_i)$ from $D$
7:     $\hat{\mathcal{B}} = \mathcal{A}(f_\theta, \mathcal{L}_{adv}, \epsilon, steps, norm)$                       ▷ adv samples
8:     $\hat{\mathcal{B}}' \leftarrow f_\theta(\hat{\mathcal{B}})$                            ▷ forward pass with adv samples
9:     $\mathcal{B}' \leftarrow f_\theta(\mathcal{B})$                              ▷ forward pass with std samples
10:     **if** $epoch > e_{warm-up}$ **then**
11:         $\hat{\mathcal{B}}'' \leftarrow f_\phi(\hat{\mathcal{B}})$                  ▷ forward pass using stable model
12:         $\mathcal{B}'' \leftarrow f_\phi(\mathcal{B})$
13:         $\mathcal{L}_{CR} = \lambda_1 \cdot D_{KL}(\hat{\mathcal{B}}' || \hat{\mathcal{B}}'') + \lambda_2 \cdot D_{KL}(\mathcal{B}' || \mathcal{B}'')$
14:         $\mathcal{L}_{FOMO} = \mathcal{L}_{adv} + \mathcal{L}_{CR}$                 ▷ refer Eq 2 & Eq 3
15:     **else**
16:         $\mathcal{L}_{FOMO} = \mathcal{L}_{adv}$
17:     Compute the gradients
18:     Update the parameters $f_\theta$

---

## A.1   IMPLEMENTATION DETAILS

We follow the standard adversarial training (AT) procedure used in previous research (Wu et al., 2020). The model was trained for a total of 200 epochs using the stochastic gradient descent (SGD) optimization algorithm with a momentum of 0.9, a weight decay of $5 \times 10^{-4}$, and an initial learning rate of 0.1. For standard AT, we reduced the learning rate by a factor of 10 at the 100[th] and 150[th] epochs, respectively. We applied standard data augmentation techniques, including random cropping with 4-pixel padding and random horizontal flipping, to the CIFAR-10 and CIFAR-100 datasets, while no data augmentation was applied to the SVHN dataset.

For training the other baseline methods (Wu et al., 2020; Chen et al., 2020; Dong et al., 2021), we used the exact same procedure and hyperparameters as specified in those methods. For the FOMO method proposed in Section 3, we began at epoch 105 ($e_{warm-up}$), a little later than the first LR decay where robust overfitting often occurs. For PreActResNet-18, we forgot a fixed $s = 3.5\%$ of the parameters in the later layers (Block-3 and Block-4) of the architecture, while for widerResNet-34-10, we forgot 5% as it has a larger capacity to memorize. Each forgetting step was followed by a relearning phase that lasted for $e_r = 5$ epochs. The relationship between $s$ and $e_r$ is studied in Section A.6. For the consolidation step, we chose a decay rate of the stable model of $\dashv_c = 0.999$. During the relearning phase, the stable model through the regularization loss ($\mathcal{L}_{CR}$), and we chose regularization strengths of $\lambda_1$ and $\lambda_2$ equal to 1. We ran the experiments for three seeds, and the average of the results is reported in the table.

**Datasets.** For our experiments, we use three datasets: CIFAR-10 (Krizhevsky et al., 2009), CIFAR-100 (Krizhevsky et al., 2009), SVHN (Netzer et al., 2011). We randomly split the original training sets for these datasets into a training set and a validation set in a 9:1 ratio. Our ablation studies and visualizations are mainly based on the CIFAR-10 dataset.

**Baseline.** We compare the results against various baseline methods such as vanilla PGD-AT (Madry et al., 2017), TRADES (Zhang et al., 2019a), KD+SWA (Chen et al., 2020), PGD-AT+TE (Dong et al., 2021), AWP Wu et al. (2020) that are proposed to mitigate robust overfitting. To assess the model's robust overfitting ability, we compare the robust test accuracy between the best-epoch and the last-epoch. The difference between the best and the final robust test accuracy is denoted as $\Delta$.

Table 5: Comparison of test robustness (%) between MLCAT and FOMO under Autoattack.

| Method | CIFAR-10 | | | CIFAR-100 | | |
|---|---|---|---|---|---|---|
| | Best | Last | $\Delta$ | Best | Last | $\Delta$ |
| $\text{MLCAT}_{LS}$ | 28.12 | 27.03 | -1.29 | 13.41 | 11.37 | -2.04 |
| $\text{MLCAT}_{WP}$ | 50.70 | 50.32 | -0.38 | 25.86 | 25.18 | -0.68 |
| FOMO | **51.37** | **51.28** | **-0.09** | **27.57** | **27.49** | **-0.08** |

Table 6: Auto-attack on FOMO using the PreActResNet-18.

| Method | CIFAR-10 | | CIFAR-100 | | Tiny-ImageNet | |
|---|---|---|---|---|---|---|
| | Best | Last | Best | Last | Best | Last |
| WA | 49.92 | 43.82 | 25.95 | 21.02 | 19.76 | 15.82 |
| KD+SWA | 49.87 | 49.74 | 26.04 | 25.99 | 19.78 | 19.76 |
| PGD-AT+TE | 50.11 | 49.14 | 26.04 | 25.13 | 18.16 | 15.88 |
| FOMO | **51.37** | **51.28** | **27.57** | **27.49** | **20.23** | **19.85** |

The results are averaged over three seeds. In addition to robust overfitting, achieving an optimal balance between natural and robust test accuracy is crucial for an effective AT method. However, there is currently no standardized method for measuring this trade-off in the adversarial trade-off literature. Therefore, we propose a trade-off measure that offers a formal approach to compare how well different methods maintain this balance. The Trade-off is measured as follows:

$$\text{Trade-off} = \frac{2 \times \mathcal{NA}_L \times \mathcal{RA}_L}{\mathcal{NA}_L + \mathcal{RA}_L} \tag{4}$$

where $\mathcal{NA}_L$ and $\mathcal{RA}_L$ stand for last-epoch natural test accuracy and robust test accuracy, respectively.

## A.2 COMPARISON WITH MLCAT

Yu et al. (2022) proposed a method called MLCAT to alleviate robust overfitting by analyzing the roles of easy and hard samples and regularizing samples with small loss values using non-robust features. In contrast, we approach this problem from the lens of active forgetting, which provides a different perspective for addressing robust overfitting.

According to the results presented in Table 5, our FOMO approach outperforms MLCAT consistently in terms of robust improvement against AutoAttack across datasets. Furthermore, FOMO exhibits lower levels of robust overfitting compared to both variants of MLCAT. $\text{MLCAT}_{LS}$ corresponds to loss scaling, and $\text{MLCAT}_{WP}$ corresponds to weight perturbation. As a result, FOMO is not only more effective than MLCAT, but also shows a reduced tendency toward overfitting. Thus, iterating periodically between forgetting, relearning, and consolidation during AT brings discernible benefits with respect to the robustness to AT.

## A.3 EXTENDED COMPARISON WITH BASELINES ON LARGE DATASETS

We conducted extensive comparisons with KD-SWA, weight averaging (WA), and model ensembling approaches on CIFAR10/100, SVHN, and Tiny-ImageNet consisting of 64x64 images. FOMO algorithm outperforms these baselines, demonstrating superior robustness and im- proved performance as shown in Table 6. Furthermore, as shown in Table 7, FOMO outperforms subspace adversarial training (Sub-AT) (Li et al., 2022). This shows the effectiveness of FOMO even on larger datasets.

## A.4 EVALUATION ACROSS PERTURBATIONS

We assess the versatility of our proposed FOMO framework by evaluating across perturbation ($\ell_2$ norm ) using PreActResNet-18 He et al. (2016) on CIFAR-10. Table 8 demonstrates the best and

Table 7: PGD-attack on FOMO against Subspace AT.

| Method | CIFAR-10 | | | CIFAR-100 | | |
|---|---|---|---|---|---|---|
| | Best | Last | $\Delta$ | Best | Last | $\Delta$ |
| Sub-AT | 52.79 | 52.31 | 0.48 | 27.50 | 27.02 | 0.48 |
| FOMO | **56.68** | **56.46** | **-0.22** | **32.07** | **31.67** | **-0.40** |

Table 8: Test robustness (%) of AT and FOMO across different datasets and threat models.

| Threat Model | Method | CIFAR-10 | | | CIFAR-100 | | |
|---|---|---|---|---|---|---|---|
| | | Best | Last | $\Delta$ | Best | Last | $\Delta$ |
| $\ell_\infty$ | PGD-AT | 52.32 | 44.44 | -7.88 | 27.22 | 20.82 | -6.4 |
| | FOMO | **56.68** | **56.46** | **-0.22** | **32.07** | **31.67** | **-0.40** |
| $\ell_2$ | PGD-AT | 69.15 | 65.93 | -3.22 | 41.33 | 35.27 | -6.06 |
| | FOMO | **72.69** | **72.28** | **-0.41** | **45.60** | **45.16** | **-0.44** |

final performance of the FOMO network compared against vanilla adversarial training (PGD-AT) with $\ell_2$ perturbation norm. We use $\epsilon$ of 128/255 and a step size of 15/255 for evaluation with $\ell_2$ perturbation. The training/test attacks are PGD-10/PGD20, respectively. The results demonstrate that FOMO significantly outperforms vanilla adversarial training across perturbations. Therefore, forgetting and relearning are more effective in mitigating robust overfitting across perturbations.

## A.5 EFFECT OF FORGETTING IN DIFFERENT LAYERS ON ROBUST OVERFITTING

Training DNNs adversarially often results in a predominant phenomenon known as robust overfitting. Current learning techniques generally analyze learning behavior by treating the network as a whole unit, which disregards the ability of individual layers to learn adversarial data distributions. We suggest that different layers possess unique capacities to learn information, and it is crucial to comprehend these patterns to develop a training scheme that mitigates robust overfitting. Therefore, we investigate the layer-wise characteristics of a network by analyzing the effect of forgetting parameters at different layers on robust forgetting. For this analysis, we use the PreActResNet18 architecture, where each block (4 blocks) is considered a layer along with an additional linear classification layer. Figure 5 demonstrates the effect of forgetting in different layers on the robust train and test accuracy.

Our analysis revealed that forgetting on early layers often results in decreased performance on robust test accuracy. Additionally, it fails to regularize the training accuracy, leading to a larger robust generalization gap. This is possibly due to the lower capacity of earlier layers to accommodate new information. Moreover, since earlier layers learn generalized features compared to later layers, forgetting them results in a loss of generalization. On the other hand, later layers have more capacity to memorize and tend to overfit the training data. Therefore, forgetting layers 3 and 4 leads to a reduced robust generalization gap, which mitigates robust overfitting. By limiting the forgetting process to the last two layers, we aim to regularize the weights in the later layers while retaining more generalized features learned in the earlier layers to facilitate robust generalization.

## A.6 STUDY THE SYMBIOTIC RELATIONSHIP BETWEEN FORGETTING AND RELEARNING

Recent advances in cognitive neuroscience have shed light on the interdependent nature of learning and forgetting, with mounting evidence indicating a symbiotic relationship between the two phenomena Bjork & Allen (1970); Bjork & Bjork (2019). In the context of DNNs, the dynamics of forgetting and relearning are of particular interest, as they have been shown to play a critical role in mitigating the deleterious effects of overfitting. Specifically, the percentage of forgotten parameters and the duration of the relearning phase ($e_r$) are important factors to consider to achieve robust overfitting.

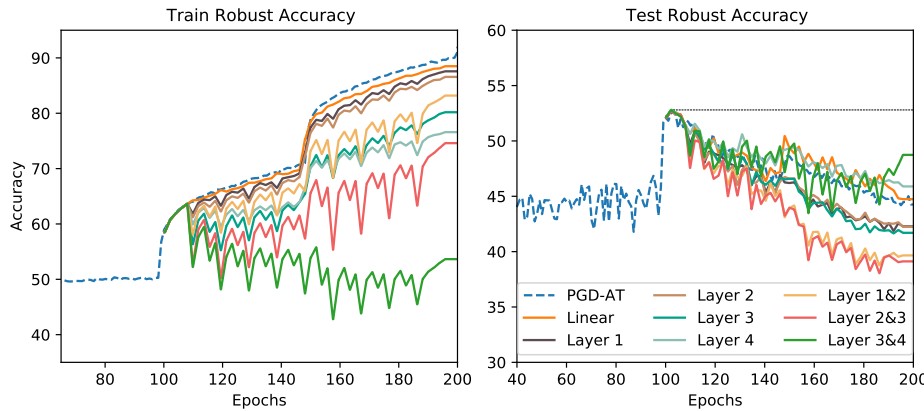

Figure 5: The impact of forgetting 50% of parameters in each layer on robust generalization using PreAct-ResNet-18 on CIFAR-10 is illustrated in the figures. The figure on the left shows the impact on train robust accuracy and the figure on the right shows the impact on test robust accuracy. It is evident from the figures that forgetting in the later layers regularizes the train robust accuracy and mitigates robust overfitting when compared to forgetting in the early layers.

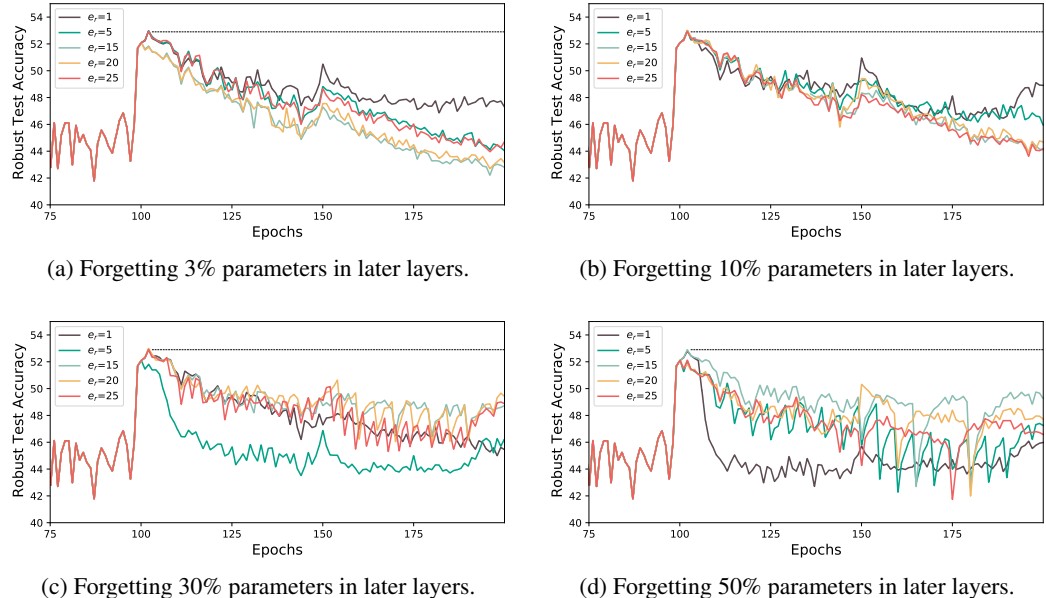

(a) Forgetting 3% parameters in later layers.

(b) Forgetting 10% parameters in later layers.

(c) Forgetting 30% parameters in later layers.

(d) Forgetting 50% parameters in later layers.

Figure 6: Study the symbiotic relationship between forgetting and relearning during adversarial training.

To investigate the impact of forgetting and relearning on overfitting, we varied the percentage of forgotten parameters (3%, 10%, 30%, and 50%) and adjusted the duration of the relearning phase over six different intervals (1, 5, 15, 20, and 25 epochs) within a total training duration of 200 epochs. Figure 6 presents the relationship between forgetting different parameters and the relearning phase during the training process.

Our results suggest that forgetting only a small percentage of parameters can effectively mitigate overfitting when combined with a relatively short relearning phase. However, forgetting many parameters with short relearning phases can make training DNNs challenging. This implies that the percentage of parameters forgotten directly correlates with the duration of the relearning phase; the higher the percentage of parameters forgotten, the longer the relearning interval required to relearn the necessary information from the previous step. Therefore, it is essential to balance the amount

Table 9: Training time per epoch on CIFAR-10 under $\varepsilon_\infty$=8/255 perturbation using PreActResNet-18 architecture.

| Method | Computation Time per Epoch (s) |
|---|---|
| PGD-AT | 132.6 |
| WA | 133.1 |
| KD+SWA | 132.6+16.5+141.7 |
| AWP | 143.8 |
| FOMO | 137.1 |

Table 10: Evaluation under CW attack on CIFAR-10 and CIFAR-100 using PreActResNet-18 architecture.

| Norm | Radius | Methods | CIFAR-10 | | CIFAR-100 | |
|---|---|---|---|---|---|---|
| | | | Best | Final | Best | Final |
| $\ell_2$ | $\frac{128}{255}$ | PGD-AT | 67.18 | 64.29 | 37.16 | 33.43 |
| | | KD+SWA | 68.87 | 68.90 | 40.56 | 40.61 |
| | | FOMO | **70.52** | **70.35** | **42.73** | **42.35** |
| $\ell_\infty$ | $\frac{8}{255}$ | PGD-AT | 47.00 | 39.96 | 22.73 | 18.11 |
| | | KD+SWA | 49.35 | 49.44 | 25.42 | 25.35 |
| | | FOMO | **52.14** | **51.95** | **26.98** | **26.61** |

of forgetting and the duration of the relearning phase to enable effective relearning and retain the critical information necessary for optimal performance.

## A.7 COMPUTATIONAL EFFICIENCY

Here, we compare the computational efficiency of FOMO with the baseline methods. To ensure a fair comparison, all methods were integrated into a universal training framework, and each test was performed on a single NVIDIA GeForce 3090 GPU. Table 9 compares the computational time per epoch of FOMO with the considered baselines. Notably, FOMO and our baselines were trained for the same number of epochs (i.e., 200 epochs for CIFAR-10/100). From Table 9, it is evident that FOMO imposes almost no additional computational cost compared to vanilla PGD-AT, with specific values of 137.1s for FOMO and 132.6s for vanilla PGD-AT per epoch. This implies that FOMO is an efficient training method in practical terms. It is important to note that KD+SWA, a formidable method designed to counter robust overfitting, comes with an increased computational cost. This arises from its approach, which entails the pretraining of both a robust and a non-robust classifier, serving as the adversarial and standard teacher, respectively. In addition, the method incorporates the process of distilling the knowledge of these teachers. Moreover, KD+SWA employs stochastic weight averaging to smooth the weights of the model, further contributing to its computational demands. We believe that this addition improves practical insight into the efficiency of FOMO and its comparison with existing methods.

## A.8 PERFORMANCE UNDER CW ATTACK

Table 10 presents the evaluation results under CW attack (Carlini & Wagner, 2017) on CIFAR-10/100 using the PreActResNet-18 architecture. The robust accuracy is assessed under CW attacks, and checkpoints with the best robust accuracy under PGD-20 attacks on the validation set are selected for comparison. FOMO consistently outperforms both baselines across different datasets and attack scenarios, demonstrating its effectiveness in enhancing robust accuracy under CW attacks. The best and final robust accuracies for FOMO are generally close, indicating that FOMO maintains its performance during training and does not suffer from significant overfitting under these attacks. These results emphasize the promising performance of FOMO in mitigating adversarial attacks, particularly under CW attack scenarios.

