# OpenReview forum: "The Effectiveness of Random Forgetting for Robust Generalization"
_ICLR.cc/2024/Conference — ICLR 2024 poster_

### Official Review · Reviewer_FBLx · 2023-10-29

**Soundness:** 2 fair
**Presentation:** 2 fair
**Contribution:** 2 fair
**Rating:** 6
**Confidence:** 3

**Summary:**

This work aims to address the generalization gap in adversarial training. The authors exploit the random forgetting to adjust the  weights of models. Three datasets and two adversarial attacks are used to evaluate the proposed method. The experimental results show that the method can improve the robust accuracy.

**Strengths:**

1. This work proposed a new method to solve the generalization gap in adversarial training.  The perspective of random forgetting is interesting.

2. The introduction to the forgetting mechanism in Methodology is clear.

**Weaknesses:**

1. The description in the caption of Figure 2 is inconsistent with the content of the image. The former states that the consolidation phase is behind the forgetting phase, while the latter expresses that theconsolidation phase is before the forgetting phase. In addition, please present more clearly in the figure what the generalized information is.

2. In Figure 1, FOMO is only compared with standard adversarial training, but not with methods that aim to reduce the generalization gap (such as AWP). This result may not appreciably represent the effectiveness of the proposed method.

3. The authors use two adversaial attacks to evaluate the proposed method, they can consider more adversarial attacks (such as L2-norm CW, DDN) to conduct a more comprehensive evaluation.

4. Figures can be clearer and more aesthetically pleasing.

**Questions:**

Please see weaknesses.

============After rebuttal============
The authors provide adequate explanations for most of my questions, so I am willing to raise the rating score to 6.

---

> ### Author Response · Authors · 2023-11-21
> **Reply to  Reviewer FBLx**
>
> > The description in the caption of Figure 2 is inconsistent with the content of the image. The former states that the consolidation phase is behind the forgetting phase, while the latter expresses that the consolidation phase is before the forgetting phase.
>
> We appreciate your valuable feedback on our manuscript. We acknowledge the inconsistency in the description of Figure 2, specifically regarding the positioning of the consolidation and forgetting phases. In the revised manuscript, we have ensured that the caption accurately reflects the chronological order depicted in Figure 2.
>
> > In Figure 1, FOMO is only compared with standard adversarial training, but not with methods that aim to reduce the generalization gap (such as AWP). This result may not appreciably represent the effectiveness of the proposed method.
>
> Figure 1 primarily serves as an illustrative aid to help readers grasp the concept of robust overfitting in adversarial training more effectively. To ensure a more comprehensive comparison, we have meticulously presented relevant details in Table 1. This table includes a dedicated section for comparing FOMO with methods like AWP and KD+SWA, which specifically aim to reduce the generalization gap.  In light of this, we have refrained from displaying the direct comparison with AWP in Figure 1 to avoid visual clutter. We value the reviewer's perspective, and in response, we will incorporate these changes in the revised version.
>
> > The authors use two adversaial attacks to evaluate the proposed method, they can consider more adversarial attacks (such as L2-norm CW, DDN) to conduct a more comprehensive evaluation.
>
> Based on reviewer's feedback, we have evaluated our proposed method under CW attack for comprehensive evaluation. Table 10  (Appendix section A8) presents the evaluation results under CW attack on CIFAR-10/100 using the PreActResNet-18 architecture. The robust accuracy is assessed under CW attacks, and checkpoints with the best robust accuracy under PGD-20 attacks on the validation set are selected for comparison. FOMO consistently outperforms both baselines across different datasets and attack scenarios, demonstrating its effectiveness in enhancing robust accuracy under CW attacks. The best and final robust accuracies for FOMO are generally close, indicating that FOMO maintains its performance during training and does not suffer from significant overfitting under these attacks. These results emphasize the promising performance of FOMO in mitigating adversarial attacks, particularly under CW attack.
>
> | Dataset   | Norm   | Radius          | Methods | CW Attack  |                 |
> |-----------|--------|-----------------|---------|------------|-----------------|
> |           |        |                 |         | **Best**   | **Final**       |
> |-----------|--------|-----------------|---------|------------|-----------------|
> | CIFAR-10  | $\ell_2$ | $\frac{128}{255}$ | PGD-AT  | 67.18      | 64.29           |
> |           |        |                 | KD+SWA  | 68.87      | 68.90           |
> |           |        |                 | FOMO    | **70.52**  | **70.35**       |
> | CIFAR-10  | $\ell_\infty$ | $\frac{8}{255}$ | PGD-AT  | 47.00      | 39.96           |
> |           |        |                 | KD+SWA  | 49.35      | 49.44           |
> |           |        |                 | FOMO    | **52.14**  | **51.95**       |
> | CIFAR-100 | $\ell_2$ | $\frac{128}{255}$ | PGD-AT  | 37.16      | 33.43           |
> |           |        |                 | KD+SWA  | 40.56      | 40.61           |
> |           |        |                 | FOMO    | **42.73**  | **42.35**       |
> | CIFAR-100 | $\ell_\infty$ | $\frac{8}{255}$ | PGD-AT  | 22.73      | 18.11           |
> |           |        |                 | KD+SWA  | 25.42      | 25.35           |
> |           |        |                 | FOMO    | **26.98**  | **26.61**       |
>
> > Figures can be clearer and more aesthetically pleasing.
>
> Thank you for your feedback on the figures. We'll work on improving their clarity and visual appeal in the final version of the paper.

---

> > ### Comment · Reviewer_FBLx · 2023-11-22
> > **Response to authors**
> >
> > Many thanks to authors for their careful responses. The authors has revised and improved the description of the figures and they add additional experimental results. I think the authors provide adequate explanations for most of my questions, so I will raise the rating score.

---

### Official Review · Reviewer_WNii · 2023-11-01

**Soundness:** 3 good
**Presentation:** 3 good
**Contribution:** 3 good
**Rating:** 6
**Confidence:** 5

**Summary:**

This paper proposed a method, FOMO, to deal with the adversarial overfiting issue. The proposed method alternates between the forgetting phase and the relearning phase.

**Strengths:**

The paper is in good structure and easy to follow.

The topic, which is to deal with adversarial overfitting, is interesting.

The method is simple yet effective.

An ablation study is provided.

**Weaknesses:**

The description of the method is too intuitive.

In Table 1, the delta, which measures the adversarial overfitting, never favors the proposed method. This cannot show that the proposed method is good at dealing with adversarial overfitting.

**Questions:**

In this paper, the author only shows the result under a combination of white box and black box attacks, i.e., Autoattack. However, this cannot show "the efficacy of FOMO against the black box and white box attacks". Standard Autoattack has 4 adversarial attacks: three white-box attacks and one black-box attack. It is possible that FOMO has a strong resistance against white-box attacks while being vulnerable to the black-box attack.

---

> ### Author Response · Authors · 2023-11-21
> **Reply to Reviewer WNii**
>
> >  The description of the method is too intuitive.
>
> We appreciate the reviewer's feedback and the opportunity to clarify the description of our method. Our FOMO approach is founded on the premise that emulating active forgetting, a phenomenon observed in the human brain, can effectively mitigate robust overfitting in Adversarial Training (AT). Our method synthesizes selective forgetting in later layers, consolidation for long-term memory, and interleaved training as mechanisms aimed at enhancing generalization.
>
> We want to emphasize that our method's development wasn't solely intuitive; it is substantiated by evidence from active forgetting studies in neuroscience, which we extensively discuss in Section 2.3 and the methodology section. Additionally, we conducted rigorous empirical evaluations across multiple datasets to validate our hypothesis. The discernible enhancements in robustness and generalization provide empirical evidence reinforcing the effectiveness of our method.
>
> We believe that while our approach may seem intuitive in its concept, its implementation and validation have been rooted in both neuroscience understanding and empirical substantiation. We remain open to further discussions or specific areas where additional clarification might be beneficial.
>
> > In Table 1, the delta, which measures the adversarial overfitting, never favors the proposed method. This cannot show that the proposed method is good at dealing with adversarial overfitting.
>
> We would like to clarify that delta alone may not be a comprehensive metric for quantifying robust overfitting, as some methods with strict regularization (such as KD+SWA) during training might exhibit lower delta values at the expense of achieving less-than-optimal the best robust accuracy. FOMO is designed to strike a balanced tradeoff between delta and best robust accuracy.
>
> Compared to KD+SWA, FOMO exhibits a notable percentage increase in the best robust accuracy: 2.08% on CIFAR-10, 2.01% on CIFAR-100, and 5.98% on SVHN, all while maintaining a minimal difference between best and final robust accuracy. These percentages highlight FOMO's effectiveness in not only mitigating adversarial overfitting but also in significantly enhancing the overall robustness of the model compared to the baseline.
>
> > In this paper, the author only shows the result under a combination of white box and black box attacks, i.e., Autoattack. However, this cannot show "the efficacy of FOMO against the black box and white box attacks". Standard Autoattack has 4 adversarial attacks: three white-box attacks and one black-box attack. It is possible that FOMO has a strong resistance against white-box attacks while being vulnerable to the black-box attack.
>
> Thank you for your question. In response to your observation, we have made revisions to the paper to explicitly address "the efficacy of FOMO against AutoAttacks." We present results on AutoAttack, a combination of white-box and black-box attacks, mirroring our considered baselines. It's crucial to note that baseline methods, such as KD+SWA and AWP, similarly lack separate results for white-box and black-box attacks. To facilitate a fair comparison, we also present the AutoAttack results. We acknowledge this and will update the paper to reflect this clarity.

---

### Official Review · Reviewer_Jeha · 2023-11-08

**Soundness:** 3 good
**Presentation:** 3 good
**Contribution:** 3 good
**Rating:** 6
**Confidence:** 3

**Summary:**

This paper addresses the challenge of robust overfitting in adversarial training of deep neural networks, which affects their generalization performance. The authors propose a new method called "Forget to Mitigate Overfitting (FOMO)," drawing inspiration from the brain's mechanism of active forgetting. FOMO operates by periodically resetting a subset of the network's weights to promote the learning of more generalizable features. The approach suggests a promising direction for enhancing neural network robustness against adversarial attacks by mitigating overfitting through controlled forgetting and relearning. Experimental results show that FOMO is a promising method to improve model robustness.

**Strengths:**

* The authors conducted comprehensive experiments to demonstrate the effectiveness of FOMO. The proposed method is effective and outperforms the existing method, according to Table 3 and other experimental results.
* The proposed method is intuitive and easy to implement.

**Weaknesses:**

* My main concern is that the proposed method seems to be heuristic and empirical. there is not enough discussion on its intuition or theoretical foundation.
* I don't think the running time and convergence analysis are well-studied in this paper, the authors may need to provide a table showing how many epochs are needed to converge and compare the running time with the existing methods.
* Minor: Please refrain from only using color to distinguish curves and bars as in Figures 3, 4, 5, and 6, as it is not friendly to readers with color blindness.
* Minor: Missing reference on robust generalization: Zhang, et al. "The limitations of adversarial training and the blind-spot attack." ICLR 2019.

**Questions:**

Please refer to the weakness part.

---

> ### Author Response · Authors · 2023-11-21
> **Reply to Reviewer Jeha**
>
> > My main concern is that the proposed method seems to be heuristic and empirical. there is not enough discussion on its intuition or theoretical foundation.
>
> We genuinely value the reviewer's insights. Our FOMO approach stems from the hypothesis that simulating active forgetting, as observed in the human brain, can effectively address robust overfitting in Adversarial Training (AT). Our method integrates selective forgetting in later layers, consolidation for long-term memory, and interleaved training as mechanisms to improve generalization. Our rationale is supported by evidence from active forgetting studies in neuroscience, which is extensively elaborated in Section 2.3 and the methodology section. Furthermore, we have empirically validated our hypothesis by conducting comprehensive evaluations across multiple datasets. The noticeable improvements in robustness and generalization serve as empirical evidence reinforcing the effectiveness of our method.
> While we acknowledge the importance of theoretical analysis, our primary focus was to showcase the practical efficacy of our proposed method. However, subsequent to our empirical findings, a theoretical exploration can further deepen our understandings of the learning mechanisms employed by FOMO. This aspect is explicitly discussed in our future work section, emphasizing the necessity of further theoretical analysis to complement our empirical findings.
>
> > I don't think the running time and convergence analysis are well-studied in this paper, the authors may need to provide a table showing how many epochs are needed to converge and compare the running time with the existing methods.
>
> We appreciate your suggestion and thus, we have conducted additional experiments and present the training time (per epoch) for several methods in our revised manuscript. To ensure a fair comparison, all methods were integrated into a universal training framework, and each test was performed on a single NVIDIA GeForce 3090 GPU. Table 9 (Appendix section A7) in the revised manuscript now includes the required information.
>
> Notably, FOMO and our baselines were trained for the same number of epochs (i.e., 200 epochs for CIFAR-10/100). From the table, it is evident that FOMO imposes nearly no extra computational cost compared to vanilla PGD-AT, with specific values being 137.1s for FOMO and 132.6s for vanilla PGD-AT per epoch. This implies that FOMO is an efficient training method in practical terms. It is important to highlight that KD+SWA, a formidable method designed to counter robust overfitting, comes with an increased computational cost. This arises from its approach, which entails the pretraining of both a robust and a non-robust classifier, serving as the adversarial and standard teacher, respectively. Additionally, the method incorporates the process of distilling knowledge from these teachers. Moreover, KD+SWA employs stochastic weight averaging to smoothen the weights of the model, further contributing to its computational demands.We believe that this addition enhances the practical insights into the efficiency of FOMO and its comparison with existing methods. We would be happy to address any remaining concern or suggestion.
>
> ### Table: Training Time per Epoch on CIFAR-10 under ε∞ = 8/255 Perturbation (PreActResNet-18)
>
> | Methods  | Training Time per Epoch (s) |
> | ------------- | ------------------------|
> | PGD-AT   | 132.6                    |
> | WA           | 133.1                    |
> | KD+SWA | 132.6 (pretraining of AT) +16.5 +141.7 |
> | AWP         | 143.8                   |
> | FOMO      | 137.1                   |
>
> > Minor: Please refrain from only using color to distinguish curves and bars as in Figures 3, 4, 5, and 6, as it is not friendly to readers with color blindness.
>
> We value your feedback. In the final version, we'll modify these figures to ensure they're accessible to readers with color vision deficiencies. Thank you for highlighting this important consideration.
>
> > Minor: Missing reference on robust generalization: Zhang, et al. "The limitations of adversarial training and the blind-spot attack." ICLR 2019.
>
> Thank you for your suggestions. We have added this in the related work under robust generalization.

---

### Meta-Review · Area_Chair_msez · 2023-12-21

**Metareview:**

The authors have made considerable efforts to address the reviewers' concerns and have substantially improved their submission. The proposed Forget to Mitigate Overfitting (FOMO) method provides an interesting approach to combat robust overfitting in adversarial training, drawing inspiration from active forgetting mechanisms in neuroscience. The authors have expanded on the methodology intuition and included a detailed discussion on the hypothesis and evidence supporting the approach, which has helped clarify the theoretical underpinnings of FOMO.

In response to concerns regarding the running time and convergence analysis, the authors have added a new table (Table 9) comparing the running time of FOMO with other existing methods, demonstrating its efficiency in practical terms. Furthermore, the authors have made changes to improve the clarity of the method description, including a revision of Figure 2 for better understanding.

The evaluation has been expanded to include additional adversarial attacks as suggested by the reviewers, with new results presented in Table 10. This comprehensive analysis further demonstrates the robustness of FOMO against various attacks. Minor changes have been integrated, including missing references and aesthetic improvements to figures for better readability, especially for individuals with color blindness.

Although Reviewer Jeha and Reviewer WNii did not actively engage during the discussion phase, the authors have still attempted to address their concerns as best as possible based on the feedback provided in the initial reviews. The authors have also responded to specific questions raised by the reviewers and have justified the effectiveness of their method in dealing with adversarial overfitting.

Considering the authors' thorough and responsive revisions, along with the strengths of the paper in its contribution to the field of adversarial training, I recommend acceptance of this paper. The concerns raised by the reviewers have been adequately addressed, and the paper now provides valuable insights and empirical evidence supporting the proposed FOMO method.

**Justification For Why Not Higher Score:**

The decision to accept the paper and not assign a higher score is based on the fact that while the authors have addressed most of the concerns raised by the reviewers, there are still some areas that could benefit from further theoretical exploration. Additionally, the lack of active engagement from certain reviewers during the discussion phase means that not all potential concerns may have been fully addressed or clarified. However, the authors have made a good-faith effort to respond to the feedback, and the improvements made to the paper are significant.

**Justification For Why Not Lower Score:**

A lower score is not warranted as the paper now includes a more comprehensive analysis, improved clarity, and addressed concerns regarding methodology and evaluation. The authors have shown that their method is not only innovative but also practical and efficient, which merits the paper's inclusion in the conference program. The additional results and revisions have provided sufficient evidence of the method's efficacy, and the paper contributes meaningfully to the field.

---

### Decision · Program_Chairs · 2024-01-16

Accept (poster)